# Enhanced Power Factor and Ultralow Lattice Thermal Conductivity Induced High Thermoelectric Performance of BiCuTeO/BiCuSeO Superlattice

**DOI:** 10.3390/ma16124318

**Published:** 2023-06-11

**Authors:** Xuewen Yang, Zhiqian Sun, Guixian Ge, Jueming Yang

**Affiliations:** 1College of Science/Xinjiang Production & Construction Corps Key Laboratory of Advanced Energy Storage Materials and Technology, Shihezi University, Shihezi 832000, China; 2Xinjiang Production and Construction Corps Key Laboratory of Oasis Town and Mountain-Basin System Ecology, Shihezi University, Shihezi 832000, China

**Keywords:** BiCuSeO, superlattice, thermoelectric materials, density functional theory

## Abstract

Based on the first-principles calculations, the electronic structure and transport properties of BiMChO (M=Cu and Ag, Ch=S, Se, and Te) superlattices have been studied. They are all semiconductors with indirect band gaps. The increased band gap and decreased band dispersion near the valence band maximum (VBM) lead to the lowest electrical conductivity and the lowest power factor for *p*-type BiAgSeO/BiCuSeO. The band gap value of BiCuTeO/BiCuSeO decreases because of the up-shifted Fermi level of BiCuTeO compared with BiCuSeO, which would lead to relatively high electrical conductivity. The converged bands near VBM can produce a large effective mass of density of states (DOS) without explicitly reducing the mobility *µ* for *p*-type BiCuTeO/BiCuSeO, which means a relatively large Seebeck coefficient. Therefore, the power factor increases by 15% compared with BiCuSeO. The up-shifted Fermi level leading to the band structure near VBM is dominated by BiCuTeO for the BiCuTeO/BiCuSeO superlattice. The similar crystal structures bring out the converged bands near VBM along the high symmetry points Γ-X and Z-R. Further studies show that BiCuTeO/BiCuSeO possesses the lowest lattice thermal conductivity among all the superlattices. These result in the *ZT* value of *p*-type BiCuTeO/BiCuSeO increasing by over 2 times compared with BiCuSeO at 700 K.

## 1. Introduction

Thermoelectric (TE) materials, which can realize the direct conversion between heat and electricity, are of great significance for sustainable development. The dimensionless figure of merit $ZT=S^{2}\sigma T/k$ is used to evaluate the performance of TE materials, where *S*, *σ*, *T,* and *k* are the Seebeck coefficient, electrical conductivity, absolute temperature, and thermal conductivity, respectively [1]. Thermal conductivity mainly consists of lattice $k_{l}$ and electronic thermal conductivity $k_{e}$. The coupling relationship that exists between *S* and *σ* makes it difficult to tune them independently [2]. The increased electrical conductivity *σ* would lead to a decrease in the Seebeck coefficient [3]. Therefore, the power factor ($S^{2}\sigma$) is used to evaluate the electrical transport properties [4]. According to the Wiedemann-Franc law, electrical thermal conductivity and electrical conductivity are also closely related [5,6].

Some oxide ceramic samples have been found to show high electronic transport properties [7,8], which have attracted a lot of attention. Therefore, the layered quaternary oxychalcogenide BiCuSeO has been widely investigated due to its low intrinsic thermal conductivity and excellent electrical transport properties [9,10]. The high phonon-phonon scattering efficiency of BiCuSeO leads to its extremely low lattice thermal conductivity [11,12]. In the past decades, many studies have been performed to enhance the TE performance of BiCuSeO. Li et al. introduced vacancies at Cu and Bi sites. They found that when the concentration of vacancies was 2.5%, the *ZT* value was 0.8 [10]. Introducing Cu vacancies could lead to decreased concentration of cations. This means the hole concentration increases. The increased hole concentration means increased electrical conductivity and a decreased Seebeck coefficient [13,14]. Ren et al. doped Pb atoms to BiCuSeO, which improved the band degeneracy near the Fermi level. The *ZT* value increased to 1.3 at 873 K [15]. Wang et al. studied the TE performance of LaCuSeO by substituting Bi atoms with La atoms. They found that the band degeneracy near the CBM was greatly enhanced, which could give rise to an obvious increase in the Seebeck coefficient for *n*-type LaCuSeO. Though the enlarged band gap value leads to a decrease in electrical conductivity, the power factor is enhanced. Thus, the *ZT* value of *n*-type LaCuSeO was 1.5 at 900 K [16]. Pan et al. dual-doped BiCuSeO by substituting 1% Cu atoms with Fe atoms and 6% Bi atoms with Pb atoms, respectively. The dual-doping effectively enhanced the power factor and decreased the thermal conductivity. The *ZT* value increased to 1.5 at 873 K [17].

Superlattice structure is an effective method to enhance the performance of TE materials [18]. The low symmetry along the *c*-axis of the superlattice could increase the anharmonicity of phonon vibration, which means there would be high phonon scattering efficiency and low lattice thermal conductivity. At the same time, the superlattice structures could also optimize the electrical transport properties, which would effectively enhance the TE performance [19]. Zhu et al. built the superlattice of Bi_2_ and Bi_2_Te_3_, which improved the TE performance by tuning the proportion of Bi_2_ and Bi_2_Te_3_. The *ZT* value was at its maximum when the proportion of Bi_2_ and Bi_2_Te_3_ was 1:5 [20]. Raju K et al. built the superlattice of SnO and PbO (SnO/PbO). They found SnO/PbO possessed a larger Seebeck coefficient and lower electrical conductivity compared with SnO and PbO. At the same time, its lattice thermal conductivity is extremely low, which led to its high *ZT* value. Compared with SnO, the *ZT* value of SnO/PbO increased by over five times. However, the maximum *ZT* value of SnO/PbO was only 0.3 at 700 K [21]. Okan Köksal et al. combined the EuO and MgO in a (EuO)_1_/(MgO)_3_(001) superlattice. Furthermore, they adjusted the lattice along the *c*-axis to improve TE performance. The *ZT|_el_* was 0.96 at 300 K, corresponding to the lattice constant of 8.4 Å along the *c*-axis [22]. Beyer et al. built a PbTe/PbSe_0.2_Te_0.8_ superlattice. The power factor of the superlattice slightly decreased because the reduced symmetry would lead to relatively low electrical conductivity. The lattice thermal conductivity can decrease by superlattice structure, which would bring out a high *ZT* value. The maximum *ZT* value of the PbTe/PbSe_0.2_Te_0.8_ superlattice was 1.3 [19]. Wang et al. built monolayer MoS_2_/MoSe_2_ superlattices and studied their TE properties. By introducing aperiodic structural defects, they decreased the phonon thermal conductivity and enhanced the *ZT* value successfully. The *ZT* value of wild ap-SL was 1.38 at 500 K [23]. In 2001, the Bi_2_Te_3_/Sb_2_Te_3_ superlattice was reported due to its extremely high *ZT* value, which was 2.4 at room temperature. Its high TE performance even exceeded the Bi_2_Te_3_ [24]. However, little research has been reported up to date on the TE performance of the BiMChO (M=Ag and Cu, Ch=S, Se, and Te) superlattice structure, though it possesses the highest *ZT* value in all oxide conductors.

In this work, several BiMChO superlattices are studied. The electronic and phonon structure and transport properties are calculated and analyzed. The power factor of *p*-type BiCuTeO/BiCuSeO (BCTO/BCSO) was found to increase by 15% compared with BiCuSeO, and its lattice thermal conductivity $k_{l}$ decreased to 25% of BiCuSeO. That means its *ZT* value would greatly increase. At the same time, the physics mechanism leading to the extremely low $k_{l}$ was analyzed by the phonon group velocity and Grüneisen parameter. There was no obvious imaginary frequency in the phonon spectrum, which indicates that the structure of BCTO/BCSO is stable. The BCTO/BCSO showed higher TE performance at high temperatures compared with Bi_2_Te_3_ series materials. Compared with chemical doping, the superlattice structure could not only increase the power factor but also decrease thermal conductivity. Due to its high TE performance, the BCTO/BCSO could be widely applied to TE refrigeration, power generation, and waste-heat recovery in the future. This work provides a new method to enhance the TE performance of BiMChO.

## 2. Computational Methods

The superlattice structures of BiMChO (M = Ag and Cu, Ch=S, Se, and Te) were built along the *c*-axis. The Perdew-Burke-Ernzerhof (PBE) of generalized gradient approximation (GGA) was chosen as the exchanged-correlation potential to optimize the crystal structures on the Vienna ab initio simulation package (VASP) [25,26,27]. The convergence accuracy was set to 10^−8^ eV. A 15 × 15 × 4 Monkhorst-Pack k-point mesh was used, and the kinetic energy cutoff was set to 450 eV The crystal structures were totally relaxed until the Hellmann-Feynman forces were less than 0.01 eV/Å. Then the WIEN 2K (version number: 21.1) package, which was based on the full-potential linearized augmented plane waves (FLAPW), was used to optimize the atomic positions, and forces as well as calculate band structures. The well-converged basis sets with R_mt_K_max_ = 7.0 were used, where R_mt_ and K_max_ were the smallest muffin-tin (MT) radius and the maximum size of reciprocal-lattice vectors, respectively. Since the GGA methods usually underestimate the band gap value, the Tran-Blaha modified Becke-Johnson (TB-mBJ) semi-local exchange potential was employed, which has been confirmed to provide a more accurate band gap than standard GGA [28,29,30]. For our calculations, spin-orbit coupling (SOC) and spin polarization were not considered.

The BoltzTraP code was used to calculate the electrical transport properties, which is based on the semi-classical Boltzmann transport theory and relaxation time approximation (RTA) [31,32,33,34]. A non-shifted mesh with 5000 k-points was used for transport calculations. For the accuracy of phonon spectra, a 3 × 3 × 1 supercell was set and the number of atoms was 144 per supercell. The density functional perturbation theory (DFPT) was employed for the phonon calculations [16,19,35]. The phonon structure was obtained by the Phonopy package. At last, the ShengBTE code was used to calculate the lattice thermal conductivity [36]. The iterative method was used and the ngrid label in CONTROL file was set to 15 × 15 × 3, which could converge well by the converging test.

## 3. Results and Discussion

### 3.1. Crystal and Electronic Structures of BiMChO (M=Cu and Ag, Ch=S, Se and Te)

As can be seen in Figure 1, the superlattice structures were built along the *c*-axis. The top half of the superlattices consists of BiAgSeO, BiCuSO, and BiCuTeO, and the bottom half consists of BiCuSeO. The low rate of lattice mismatch means high physical stability. Then the lattice constants were optimized using the PBE exchange-correlation potentials. After the optimization, the lattice constants of BiCuSeO were slightly larger than the experiment values (see Table 1). This is mainly due to the lattice constants being usually overestimated by the PBE exchange-correlation potentials [37]. The larger atomic radii of Ag and Te atoms lead to the lattice constants of the superlattice structures of BiAgSeO/BiCuSeO (BASO/BCSO) and BCTO/BCSO being slightly larger than that of BiCuSeO.

Figure 2 gives the band structures of the superlattices of BiMChO. The k-points of the first Brillouin zones of all the superlattices were automatically generated by the software XcrySDen (verison number: 1.6.2) along the high symmetry points of Γ-X-M-Γ-Z-R-A-Z The TB-mBJ exchange-correlation potential was employed for the calculations, which has been confirmed to provide a more accurate band gap than the PBE. The similar crystal structures result in the similar band structures in Figure 2. BiCuSeO and its superlattices are indirect band-gap semiconductors. However, the band dispersion and convergence near the Fermi level are different, which leads to different electrical transport properties. This is because the band structure near the Fermi level can greatly affect TE performance. There is no obvious band convergence occurring for *p*-type BiCuSeO near VBM (Figure 2a). However, band convergence has been clearly observed between high symmetry points Γ-X and between Z-R for superlattice BCTO/BCSO near VBM (Figure 2d). The converged bands near VBM would lead to a relatively high Seebeck coefficient for *p*-type BiCuTeO/BiCuSeO. The origin of the band convergence for BCTO/BCSO can be explained by Appendix A, which gives the schematic diagram of the crystal structure and its high symmetry points. The structures are the same along the directions of Γ-X and Z-R in Appendix A. Therefore, the band structures are similar along the Γ-X and Z-R directions. It can be further found that the band structure of BCTO/BCSO near VBM is mainly determined by BiCuTeO, as can be seen from Figure 2a,d and Appendix A. This is because the Fermi level of BiCuTeO (7.32 eV) is higher than that of BiCuSeO (5.63 eV). However, the band structure near CBM is mainly determined by BiCuSeO, as can be seen by comparing Figure 2a,d and Appendix A. Therefore, the increased energy level of VBM would lead to a decreased value of the band gap of BCTO/BCSO. This would increase electrical conductivity and the power factor for *p*-type BCTO/BCSO.

Figure 2b demonstrates that an increased band gap and decreased band dispersion near VBM would lead to an enormous decrease in electrical conductivity for *p*-type BASO/BCSO compared with BiCuSeO. Therefore, BASO/BCSO possesses the worst power factor. Figure 2c shows that BiCuSO/BiCuSeO has a similar band structure as BiCuSeO near VBM. Nevertheless, the band gap value of BCSO/BCSO is higher than that of BiCuSeO. Therefore, the electrical conductivity of BCSO/BCSO is lower than that of BiCuSeO. This leads to a relatively worse power factor for BCSO/BCSO.

Figure 3 shows that the band gap values of superlattices are between the two composed materials. The band gap value of BiCuTeO is the smallest compared with BiAgSeO and BiCuSO. Therefore, the band gap value of BCTO/BCSO is relatively smaller, which may mean that the electrical conductivity of BCTO/BCSO is the largest. This could result in the largest power factor for BCTO/BCSO. The largest band gap of BiAgSeO leads to the relatively large band gap of BASO/BCSO. This may give rise to the lowest electrical conductivity and the lowest power factor compared with all the other superlattices.

The electronic localization functions (ELFs) were calculated by the VASP package to analyze the bonding characteristics of superlattices of BiMChO in Figure 4. The isosurface value was set to 0.25 Å^−3^. There is almost no difference in the bonding properties of BiCuSeO, which is located at the bottom parts of the superlattices (see Figure 4, black circles). However, the bonding characteristics of BiAgSeO, BiCuSO, and BiCuTeO are different. The covalent bonding states between Te-Bi atoms are more obvious than that of S-Bi, and Se-Bi atoms (see Figure 4, blue circles). This is due to the smallest electronegativity of Te atoms. The large atomic radius of the Te atom means a weak binding state with the Cu atom. The relatively weak intra-layer interaction leads to relatively strong inter-layer interaction between Te and Bi atoms. The stronger intra-layer binding state between S and Cu atoms leads to relatively weak inter-layer interaction between S and Bi atoms.

The high peak value of the effective density of states (DOS) near the Fermi level represents a high Seebeck coefficient [38]. Then, the DOS of superlattices of BiMChO was calculated. As can be seen in Figure 5, the dominant contribution atoms in the superlattices near the VBM and CBM are Cu and Bi atoms. The peak value of DOS near the VBM of BCSO/BCSO is very close to BiCuSeO. They have a close Seebeck coefficient. The peak value of DOS near the VBM of BASO/BCSO and BCTO/BCSO is smaller compared with BiCuSeO, which means a lower Seebeck coefficient. As for the CBM, the peak value of DOS of BCTO/BCSO is obviously smaller than the others. The Seebeck coefficient of *n*-type BCTO/BCSO is the smallest at the same carrier concentration.

The electrical transport properties are determined by the band structures near the Fermi level [39]. As shown in Figure 6, the band decomposed charge densities of BiMChO’s superlattices were calculated by the VASP package to analyze the electrical conductivity. The top half of Figure 6 gives the band decomposed charge densities of BiMChO’s superlattices near the CBM. BiAgSeO and BiCuSO play a dominant role in the electrical conductivity in the superlattices of BASO/BCSO and BCSO/BCSO. BiCuSeO plays a dominant role in the BiCuTeO/BiCuSeO superlattice. At the same time, the most obvious conductive pathway (see Figure 6, red circles) is formed between Bi and Cu atoms in BCTO/BCSO, which indicates that the electrical conductivity of *n*-type BCTO/BCSO is the largest one. The bottom part of Figure 6 gives the band decomposed charge densities near the VBM. The top and bottom parts both contribute to the electrical conductivity in BCSO/BCSO and BCTO/BCSO. However, BiCuSeO plays a major role in electrical conductivity in the BASO/BCSO. This leads to relatively low electrical conductivity. All the atoms’ contributions near the Fermi level are consistent with DOS in Figure 5.

### 3.2. Transport Properties of BiMChO (M=Cu and Ag, Ch=S, Se and Te)

The electrical transport properties of BiCuSeO and its superlattices were calculated using the BoltzTraP code, which is based on the semi-classical Boltzmann transport theory. The rigid band approximation (RBA) was used to simulate doping. As can be seen in Figure 7, the carrier concentration dependent σ/τ, Seebeck coefficient, and $S^{2}\sigma /\tau$ under 300 K and 700 K are studied.

Figure 7a–d show the carrier concentration dependent σ/τ of BiCuSeO and its superlattices under 300 and 700 K. The large band dispersion near the CBM leads to the large σ/τ for *n*-type BiCuSeO. Its maximum value was 6.76 × 10^19^ Sm^−1^s^−1^ under 300 K, corresponding to the carrier concentration of −1 × 10^21^ cm^−3^. However, for *p*-type σ/τ, BCTO/BCSO is the largest one in all the superlattices. This can be confirmed by the conductive pathway between Bi-Cu atoms in Figure 6, red circles. The maximum value of σ/τ was 2.98 × 10^19^ Sm^−1^s^−1^ under 300 K, corresponding to a carrier concentration of 1 × 10^21^ cm^−3^. Due to the coupling relationship between σ/τ and the Seebeck coefficient, the high σ/τ leads to a small Seebeck coefficient for *p*-type BCTO/BCSO. There is only BiCuSeO at the bottom half of the superlattice, which contributes to the electronic transport near the VBM. This results in extremely low electrical conductivity for the *p*-type BASO/BCSO.

Figure 7e–h show the carrier concentration-dependent Seebeck coefficient of BiCuSeO and its superlattices under 300 and 700 K. The peak value of DOS near the VBM is close to BiCuSeO and BCSO/BCSO, which was 6.9 states/eV (see Figure 5). This leads to a similar Seebeck coefficient for *p*-type BiCuSeO and BCSO/BCSO, both of which were 200 $\mu VK^{-1}$ under 700 K, corresponding to a carrier concentration of 1 × 10^21^ cm^−3^. The low peak value of effective DOS near the VBM of BASO/BCSO and the slowly increased peak value near the VBM of BCTO/BCSO lead to their low Seebeck coefficient. The rapidly increased peak value of DOS near the Fermi level means a relatively high Seebeck coefficient. The peak value near the CBM of BiCuSeO, BASO/BCSO, and BCSO/BCSO are close, which is about 6.7 states/eV (see Figure 5). Their Seebeck coefficient was −95 $\mu VK^{-1}$ under 700 K, corresponding to a carrier concentration of −1 × 10^21^ cm^−3^. The first peak value of DOS near the CBM for BCTO/BCSO is only 3.7 states/eV (see Figure 5). This leads to the low Seebeck coefficient for *n*-type BCTO/BCSO. Its Seebeck coefficient was −77 $\mu VK^{-1}$ under 700 K, corresponding to a carrier concentration of −1 × 10^21^ cm^−3^.

The high σ/τ and large Seebeck coefficient mean a large power factor. Therefore, the power factor relative to the relaxation time, $S^{2}\sigma /\tau$, is calculated in Figure 7i–l. Due to the smaller σ/τ and close Seebeck coefficient compared with BiCuSeO for *n*-type BiAgSeO/BiCuSeO and BiCuSO/BiCuSeO, their values of $S^{2}\sigma /\tau$ are low. The σ/τ and Seebeck coefficient of *n*-type BCTO/BCSO are both smaller than that of *n*-type BiCuSeO, which results in its small value of $S^{2}\sigma /\tau$. As for the *p*-type BCTO/BCSO, its extremely higher σ/τ and slightly smaller Seebeck coefficient compared with *p*-type BiCuSeO lead to relatively large $S^{2}\sigma /\tau$. The maximum value of $S^{2}\sigma /\tau$ of *p*-type BCTO/BCSO (BiCuSeO) was 5.7 × 10^11^ WK^−2^m^−1^s^−1^ (4.9 × 10^11^ WK^−2^m^−1^s^−1^) at 700 K, corresponding to the carrier concentration of 0.5 × 10^21^ cm^−3^ (1 × 10^21^ cm^−3^).

The phonon spectra were calculated using the Phonopy and VASP packages to analyze the phonon transport properties and stability. In order to ensure the accuracy of calculations, a 3 × 3 × 1 supercell was set. As can be seen in Figure 8, there is no obvious imaginary frequency for BCSO/BCSO and BCTO/BCSO, which means high stability. However, there is more imaginary frequency around the high symmetry points for BASO/BCSO. The poor stability for BASO/BCSO limits its application. The frequency gaps for BASO/BCSO and BCTO/BCSO are 1.72 THz and 1.65 THz, respectively.

The Grüneisen parameters and phonon group velocities of the superlattices are calculated to further understand the phonon transport properties. The large extremum of Grüneisen parameters and small group velocities means high anharmonicity [26]. The strong anharmonicity leads to the high efficiency of phonon scattering and low lattice thermal conductivity.

Figure 9 shows the Grüneisen parameters dependent on the vibration frequency of phonons of the superlattices. The extremums of Grüneisen parameters of BASO/BCSO, BCSO/BCSO, and BCTO/BCSO are 16.9, 4.9, and 19.8, respectively. The large extremum of Grüneisen parameters of BCTO/BCSO means high anharmonicity. For three superlattices, the ZA branches play a dominant role in the phonon vibrations.

Figure 10 shows phonon group velocities of BiMChO and its superlattices, which are dependent on the vibration frequency of phonons. The top half consists of the phonon group velocities along the *a*-direction. The phonon group velocities of BCTO/BCSO are 3.63 km/s and 3.68 km/s for ZA and LA branches, respectively, which is slightly smaller compared with BASO/BCSO and BCSO/BCSO. The bottom half consists of the phonon group velocities along the *c*-direction. The phonon group velocity of BiCuSeO is 3.43 km/s for the LA branch, which is much larger compared with that of BCTO/BCSO (1.44 km/s). The low phonon group velocities along the *c*-direction mean that BCTO/BCSO has low lattice thermal conductivity.

At last, the temperature-dependent lattice thermal conductivity ($k_{l}$) of BiMChO and its superlattices are calculated using the ShengBTE code, which is based on the Boltzmann transport theory. As can be seen in Figure 11, the $k_{l}$ of BiCuSeO is the largest one compared with all the superlattices. At 300 K and 800 K, the $k_{l}$ of BiCuSeO are 0.85 W/m K and 0.33 W/m K, respectively. It is very close to experiment [7]. The $k_{l}$ of the superlattices of BiMChO decreased. The $k_{l}$ of BCTO/BCSO is extremely low compared with BiCuSeO. Its $k_{l}$ is 0.21 W/mK and 0.08 W/mK at 300 K and 800 K, which is only 25% of BiCuSeO.

By the above analysis of electronic and phonon transport properties of superlattices, it can be seen that the σ/τ of *p*-type BCTO/BCSO increased by two times compared with BiCuSeO. Though its Seebeck coefficient decreased slightly, its $S^{2}\sigma /\tau$ increased by 15%. At the same time, the $k_{l}$ of BCTO/BCSO decreased to 25% of BiCuSeO, which is only 0.09 W/mK at 700 K. We assume that the relaxation time of BCTO/BCSO is the same with BiCuSeO. The increased σ/τ of BiCuTeO/BiCuSeO leads to the increase of $k_{e}$. According to the work on BiCuSeO from Wang et al. [12], the $k_{e}$ of BCTO/BCSO is about 0.43 W/mK at 700 K, which is twice that of BiCuSeO. The relatively low thermal conductivity and large power factor lead to its high *ZT* value. The maximum *ZT* value of *p*-type BCTO/BCSO was 1.3, which increased by over 2 times compared with BiCuSeO.

## 4. Conclusions

The electronic structures and transport properties of superlattices of BiMChO (M=Ag and Cu, Ch=S, Se, and Te) were studied by first-principles calculation and Boltzmann transport theory. The following aspects are summarized. (1) The decreased band gap and large band dispersion near the VBM lead to the large σ/τ for *p*-type BCTO/BCSO, which is two times larger compared with BiCuSeO. Though the small peak value of effective DOS near the VBM results in its low Seebeck coefficient, the $S^{2}\sigma /\tau$ of *p*-type BCTO/BCSO increased by 15% compared with BiCuSeO. (2) The relatively larger Grüneisen parameter of BCTO/BCSO means high inharmonicity for phonon vibration and high efficiency for phonon scattering. At the same time, its low phonon group velocity along the *c*-direction also leads to low lattice thermal conductivity. The $k_{l}$ of BCTO/BCSO is 0.09 W/mK at 700 K, which is much smaller than that of BiCuSeO. (3) The *ZT* value of *p*-type BCTO/BCSO increases by over 2 times and arrived at 1.3 at 700 K. This work realizes the increase of power factor and decrease of lattice thermal conductivity simultaneously. The detailed analyses of electronic and phonon properties provide deep insight into the TE properties of superlattices. Though the construction of superlattice is relatively difficult in experiments compared with doping and some other common synthetic methods, we hope that our theoretical study may stimulate the research on the application of superlattice structures of BiMChO.

## Figures and Tables

**Figure 1 materials-16-04318-f001:**
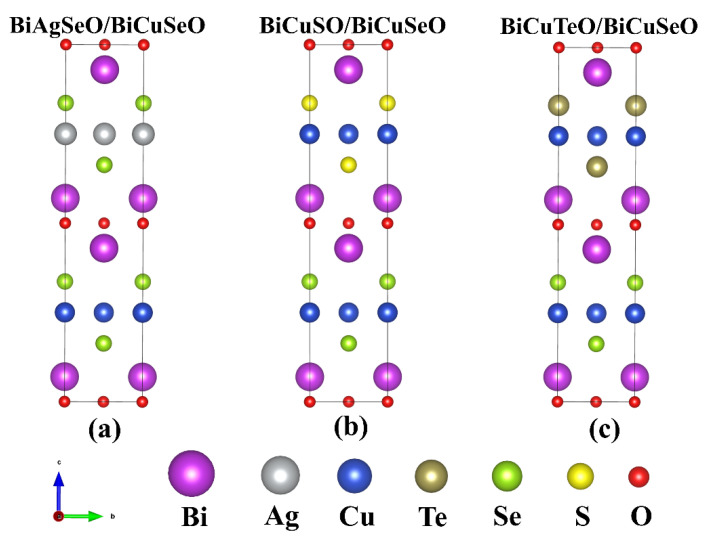
The crystal structures of BiMChO (M=Cu and Ag, Ch=S, Se, and Te). (**a**) BASO/BCSO, (**b**) BCSO/BCSO, (**c**) BCTO/BCSO. The top half of the crystals consists of BiAgSeO, BiCuSO, and BiCuTeO. The bottom half consists of all BiCuSeO.

**Figure 2 materials-16-04318-f002:**
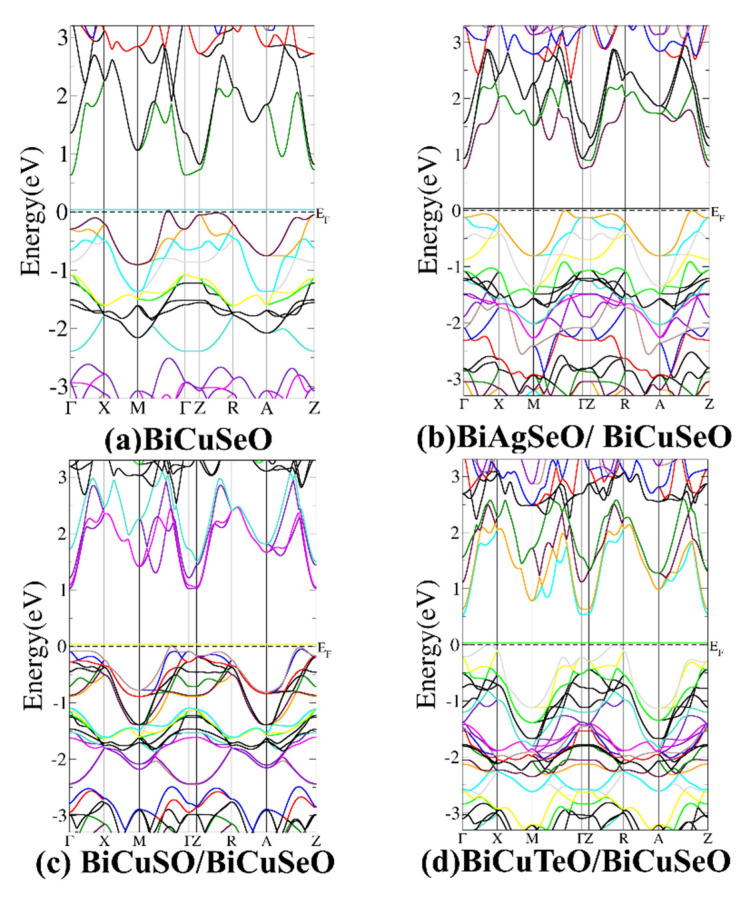
Band structures of BiCuSeO and its superlattices. (**a**–**d**) are BiCuSeO, BASO/BCSO, BCSO/BCSO, and BCTO/BCSO, respectively. The VBM of BCTO/BCSO shows high degeneracy.

**Figure 3 materials-16-04318-f003:**
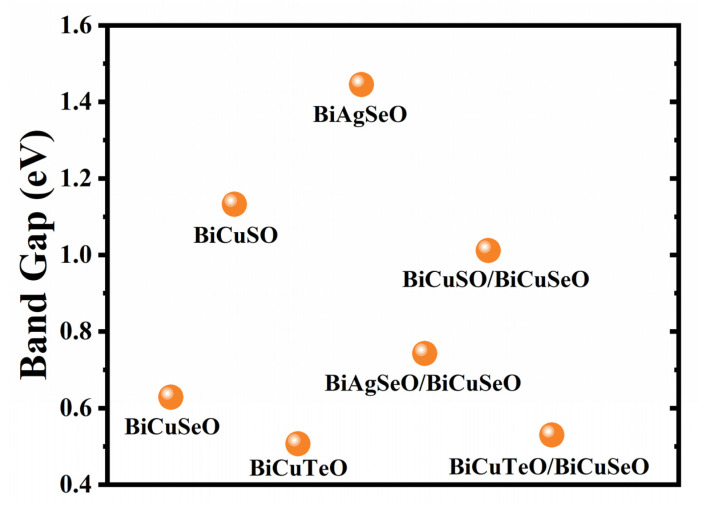
The band gap values of BiMChO and their superlattices.

**Figure 4 materials-16-04318-f004:**
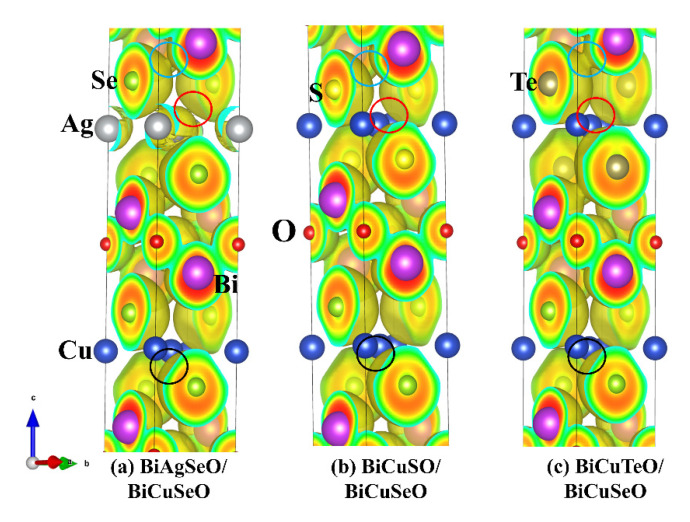
The ELFs of superlattices of BiMChO, the isosurface value was set to 0.25 Å^−3^. (**a**–**c**) are BASO/BCSO, BCSO/BCSO, and BCTO/BCSO, respectively.

**Figure 5 materials-16-04318-f005:**
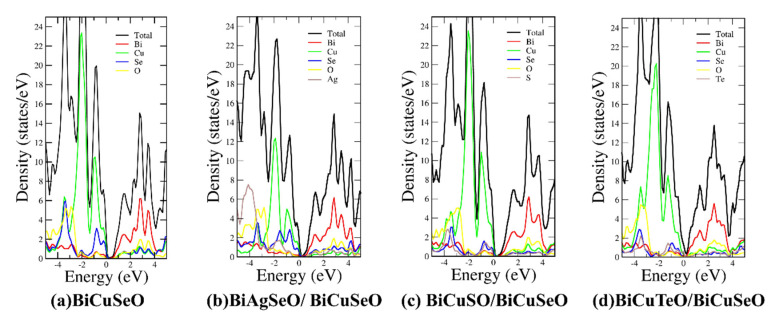
The DOS of BiMChO and its superlattices. (**a**–**d**) are BiCuSeO, BASO/BCSO, BCSO/BCSO, and BCTO/BCSO, respectively.

**Figure 6 materials-16-04318-f006:**
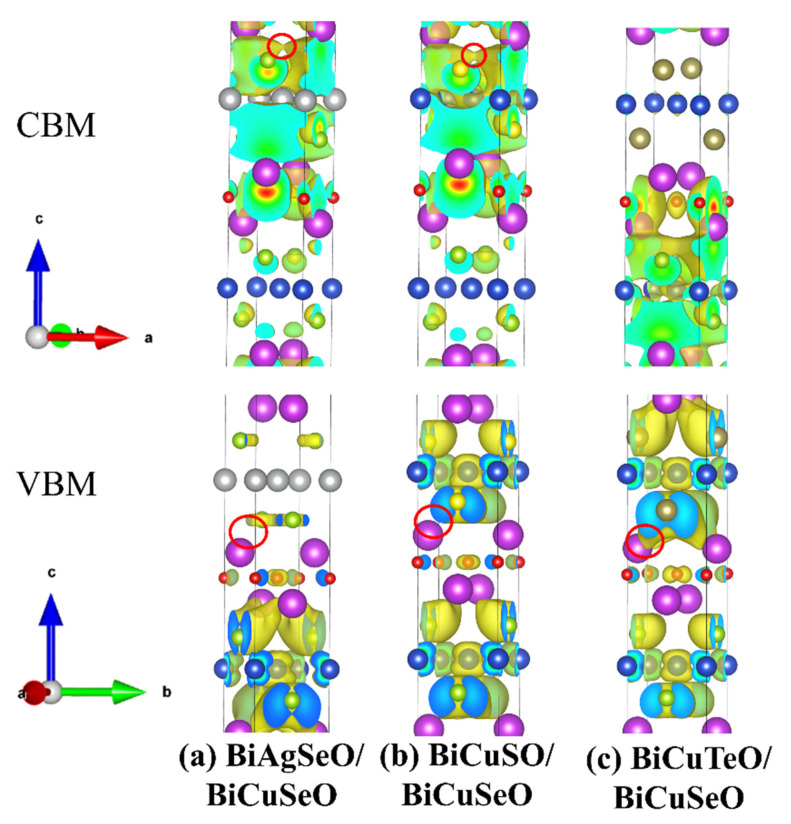
Band decomposed charge densities of the superlattices. The isosurface value is 0.001 Å^−3^. (**a**–**c**) are BASO/BCSO, BCSO/BCSO, and BCTO/BCSO, respectively.

**Figure 7 materials-16-04318-f007:**
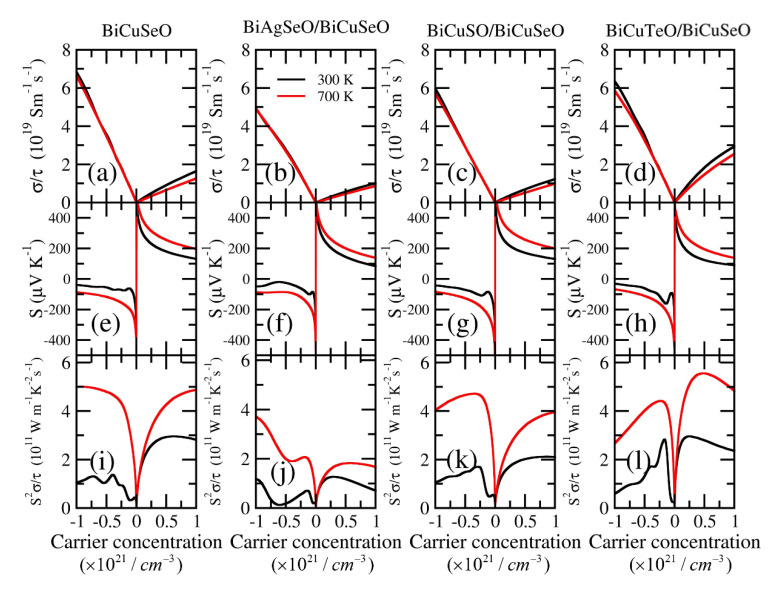
The carrier concentration-dependent electronic transport properties of BiMChO and its superlattices at 300 K (black line) and 700 K (red line). (**a**–**d**) are the electrical conductivities relative to the relaxation time σ/τ (unit in 10^19^ Sm^−1^s^−1^); (**e**–**h**) are the Seebeck coefficient *S* (unit in μVK^−1^); (**i**–**l**) are the power factor with respect to the relaxation time $S^{2}\sigma /\tau$ (unit in 10^11^ WK^−2^m^−1^s^−1^).

**Figure 8 materials-16-04318-f008:**
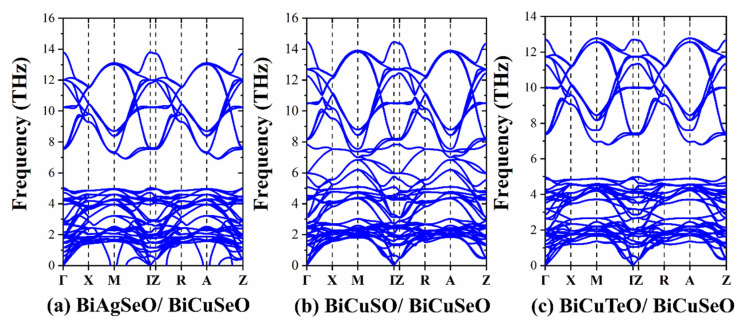
The phonon spectra (**a**–**c**) for BiAgSeO/BiCuSeO, BiCuSO/BiCuSeO, and BiCuTeO/BiCuSeO.

**Figure 9 materials-16-04318-f009:**
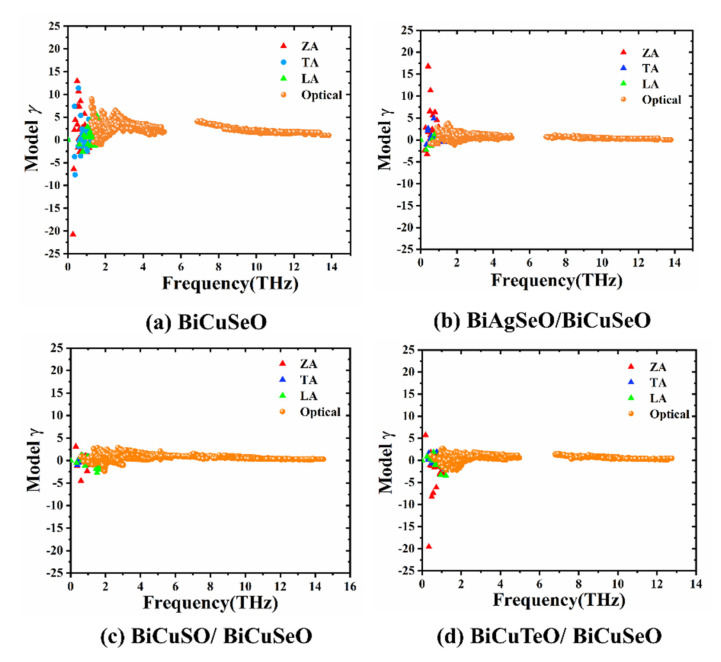
The phonon vibration frequency-dependent Grüneisen parameter of superlattices of BiMChO. (**a**–**d**) are BiCuSeO, BASO/BCSO, BCSO/BCSO and BCTO/BCSO.

**Figure 10 materials-16-04318-f010:**
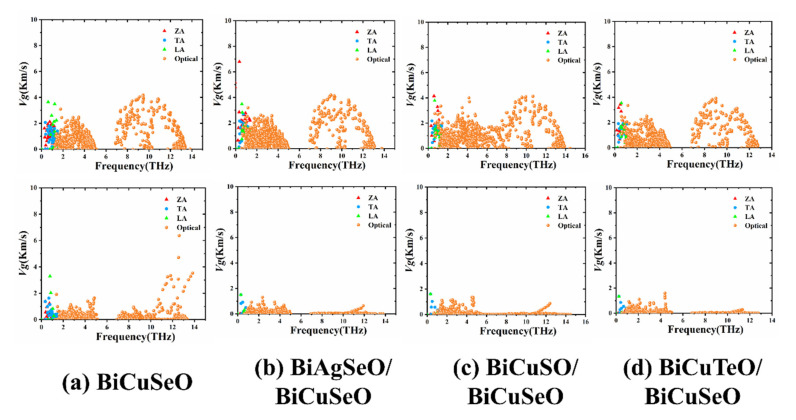
The phonon vibration frequency-dependent phonon group velocities of BiMChO and its superlattices. (**a**–**d**) are BiCuSeO, BASO/BCSO, BCSO/BCSO and BCTO/BCSO.

**Figure 11 materials-16-04318-f011:**
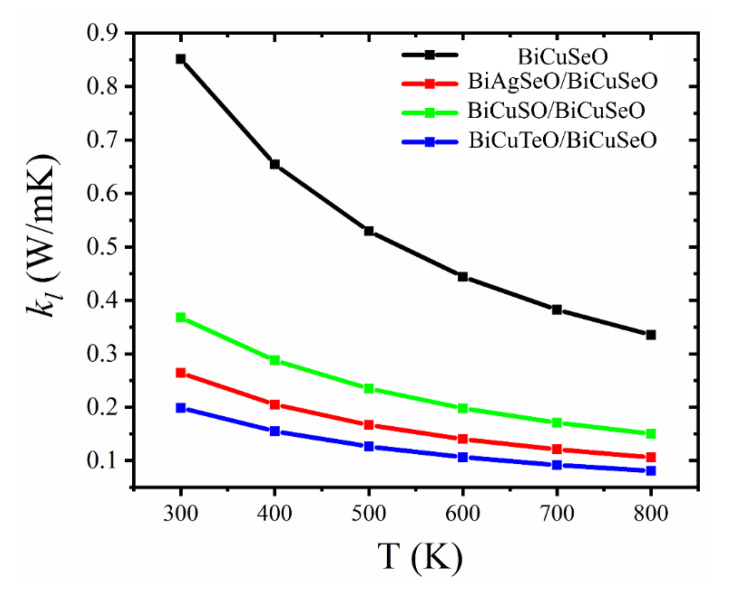
The temperature-dependent lattice thermal conductivity ($k_{l}$) of BiCuSeO and its superlattices.

**Table 1 materials-16-04318-t001:** The calculated crystal parameters and band gap values of BiMChO (M=Cu and Ag, Ch=S, Se, and Te). Due to the properties of PBE, the lattice parameters of BiCuSeO are larger compared to the experiment, and the band gap values are smaller compared to the experimental results.

	a-, b-axis (Å)	c-axis (Å)	Band Gap (eV)
BiCuSeO	3.95	9.09	0.63
BiCuSeO(expt) [12]	3.92	8.91	0.8
BASO/BCSO	3.99	18.59	0.74
BCSO/BCSO	3.91	17.75	1.01
BCTO/BCSO	4.00	18.70	0.53

## Data Availability

Data availability is not applicable to this article as no new data were created or analyzed in this study.

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
