# Peer review of "Enhanced Power Factor and Ultralow Lattice Thermal Conductivity Induced High Thermoelectric Performance of BiCuTeO/BiCuSeO Superlattice"

_materials, 2023, doi:10.3390/ma16124318_

Round 1

Reviewer 1 Report

The English language appears to be of moderate quality. While the overall content is understandable, there are some areas where grammar, sentence structure, and clarity of expression could improve the article's readability and flow. A thorough editing process is recommended to address these minor language issues and ensure a smoother presentation of the research findings.

Author Response

Thank you for your comments and revised suggestions. We have tried our best to revised. Our response and revised manuscript are attached. 

Reviewer 2 Report

The manuscript describes a first principles computational study, based on density-functional theory, of the ability of superlattice BiCuTeO/BiCuSeO structures to support thermoelectric behavior. The work considers the stability of these structure, via an analysis of they phonon characteristics, their electronic properties (band gap and chemical bond nature), and their electric transport properties. It is shown that the creation of superlattice structures increases the band gap and enhances the thermoelectric performance of these materials compared to BiCuSeO. A high degree of anharmonicity is also shown to appear in the the superlattices, along with an enhanced photon scattering efficiency.

The work is competently carried out. Structural and vibrational properties are calculated using the PBE functional, whereas, for the band gap and the electronic properties, the authors use the Tran-Blaha functional. This demonstrates that a lot of thinking went into making sure that the data presented are reliable and physically meaningful. The main findings of the work are well supported by the calculations, and they are important for the study of new thermoelectric materials with non-trivial structure.

The paper is scientifically sound and publishable in my opinion. However the quality of the English should be improved substantially, as some sentences are quite obscure and there are a lot of language errors and typos.

The manuscript requires extensive revision. Although the paper is readable, there are sentences that are difficult to make sense of, along with a lot of grammar/orthography errors (e.g., line 125: "Brillion" should be "Brillouin"; line 306: "analyzation" should be "analysis"; etc.).

Author Response

(The authors gave the same response as above.)

Reviewer 3 Report

Referee Report

On the paper “ Enhanced power factor and ultralow lattice thermal conductivity induced high thermoelectric performance of BiCuTeO/BiCuSeO superlattice “ (materials-2407671) by the authors Xuewen Yang, Qingcheng Dong, Zhiqian Sun, Guixian Ge and Jueming Yang submitted to the Materials

This is interesting theoretical paper. Based on the first-principles calculations, the electronic structure and transport properties of BiMChO (M=Cu and Ag, Ch=S, Se and Te) superlattices are studied. It was established that the band gap value of BiCuTeO/BiCuSeO was decreased because of the up-shifted Fermi level of BiCuTeO compared with BiCuSeO, which would lead to relatively high electrical conductivity. The power factor increased for 15% compared with BiCuSeO. The up-shifted Fermi level leaded to the band structure near valence band maximum was dominated by BiCuTeO for the BiCuTeO/BiCuSeO superlattice. It was shown that BiCuTeO/BiCuSeO possessed the lowest lattice thermal conductivity among all the superlattices. In my opinion, the obtained theoretical results are worthy of attention. However, a few points should be added and improved. I think that this paper can be published only after corresponding additions and corrections:

1.    I understand the choice of the object of study. These are the doped oxide ceramic samples which have good electronic properties. I fully agree with the authors that: “ The layered quaternary oxychalcogenide BiCuSeO has been widely investigated due to its low intrinsic thermal conductivity and excellent electrical transport properties [7, 8]. ”. However, there are different classes of the oxide ceramic samples which have excellent electronic properties and one of them is the ferrites:

(1). V.A. Turchenko, A.V. Trukhanov, I.A. Bobrikov, S.V. Trukhanov, A.M. Balagurov, Study of the crystalline and magnetic structures of BaFe11.4Al0.6O19 in a wide temperature range, J. Surf. Investig. 9 (2015) 17-23. https://doi.org/10.1134/S1027451015010176.

(2). M.V. Zdorovets, A.L. Kozlovskiy, D.I. Shlimas, D.B. Borgekov, Phase transformations in FeCo – Fe2CoO4/Co3O4-spinel nanostructures as a result of thermal annealing and their practical application, J. Mater. Sci.: Mater. Electron. 32 (2021) 16694-16705. https://doi.org/10.1007/s10854-021-06226-5.

The authors should note this information in Introduction.

2.    For the metals oxides the stoichiometry is particularly important. The deviation of the concentration of the original cations from a given value can lead to a change in the charge state of the transition metal cations, which in turn will greatly change the magnetic and electrical parameters:

(3). S.V. Trukhanov, D.P. Kozlenko, A.V. Trukhanov, High hydrostatic pressure effect on magnetic state of anion-deficient La0.70Sr0.30MnOx perovskite manganites, J. Magn. Magn. Mater. 320 (2008) e88-e91. https://doi.org/10.1016/j.jmmm.2008.02.021.

(4). A. Kozlovskiy, K. Egizbek, M.V. Zdorovets, M. Ibragimova, A. Shumskaya, A.A. Rogachev, Z.V. Ignatovich, K. Kadyrzhanov, Evaluation of the efficiency of detection and capture of manganese in aqueous solutions of FeCeOx nanocomposites doped with Nb2O5, Sensors 20 (2020) 4851. https://doi.org/10.3390/s20174851.

Effect of the oxygen stoichiometry on the unit cell parameters and charge state of cations for obtained samples should be given in more details and discussed in the context of their relationship with electronic properties in 3. Results and discussion.

3.    The presented 4 papers should be inserted in References.

The paper should be sent to me for the second analysis after the moderate revisions.

Minor editing of English language required

Author Response

(The authors gave the same response as above.)

Reviewer 4 Report

BiCuSeO oxyselenides are promising materials with high thermoelectric performance. The study described in this manuscript contains a comprehensive theoretical investigation of superlattice structures based on BiMChO materials with the aim of identifying structures that could improve the properties of these materials. These materials have been extensively studied in recent years, and the present study makes a valuable contribution to the field.

In this manuscript, the computational methodology for determining the different properties of the four proposed structures is clearly presented, mentioning all the different software packages and options. The results are discussed in detail to determine the structure with the best thermoelectric performance, presenting the results in the form of graphs or schematics.

The discussion is rich in analysis of results and provides details and supporting literature, but the text is very dense and difficult to read, and the relationship between text and graphs is difficult to follow and discern.

The constant repetition of the chemical composition of materials in the structures interrupts the flow of reading. The use of acronyms to identify these structures would be very helpful and greatly improve the reader interest.

The recommendation is to accept the manuscript, but a revision should be made to improve the clarity and readability of the text.

Author Response

(The authors gave the same response as above.)

Round 2

Reviewer 3 Report

Referee Report

On the paper “ Enhanced power factor and ultralow lattice thermal conductivity induced high thermoelectric performance of BiCuTeO/BiCuSeO superlattice “ (materials-2407671-v2) by the authors Xuewen Yang, Qingcheng Dong, Zhiqian Sun, Guixian Ge and Jueming Yang submitted to the Materials

This paper has been well corrected and it can be recommended.

Minor editing of English language required